

# UV measurements at Marambio and Ushuaia during 2000–2010

Kaisa Lakkala[1,2], Alberto Redondas[3], Outi Meinander[1], Laura Thölix[1], Britta Hamari[1], Antonio Fernando Almansa[3], Virgilio Carreno[3], Guillermo Deferrari[4,5], Hector Ochoa[6], Germar Bernhard[7], Ricardo Sanchez[8], and Gerardus de Leeuw[1]

[1]Finnish Meteorological Institute, Climate Research, Helsinki, Finland
[2]Finnish Meteorological Institute, Arctic Research, Sodankylä, Finland
[3]Izaña Atmospheric Research Center, Agencia Estatal de Meteorología, Tenerife, Spain
[4]Centro Austral de Investigaciones Cientificas (CADIC/CONICET), Ushuaia, Argentina
[5]Universidad Nacional de Tierra del Fuefo, Ushuaia, Argentina
[6]Dirección Nacional del Antártico-Instituto Antártico Argentino, Buenos Aires, Argentina
[7]Biospherical Instruments, Inc., San Diego, U.S.A
[8]Servicio Meteorológico Nacional, Argentina

*Correspondence to:* Kaisa Lakkala (kaisa.lakkala@fmi.fi)

**Abstract.** Solar ultraviolet (UV) irradiances were measured with NILU-UV multichannel radiometers at Ushuaia (54°S) and Marambio (64°S) between 2000 and 2013. The measurements were part of the Antarctic NILU-UV network, which was started in cooperation between Spain, Argentina and Finland. The erythemally weighted UV irradiance time series of both stations were analyzed for the first time in this study. The quality assurance procedures included a traveling refence instrument to

5    transfer the irradiance scale to the stations. The time series were homogenized and high quality measurements were available for the period 2000–2010. During this period UV indices of 11 or more were measured on 5 and 35 days at Marambio and Ushuaia, correspondingly. At Marambio, the peak daily maximum UV index of 12 and dailydoses of around 7 $kJ/m^2$ were measured in November 2007. Typically the highest UV daily doses were at both stations around 6 $kJ/m^2$, and they occurred on time periods, when the station was inside the polar vortex with very low total ozone amount. At both stations, dailydoses of

10   late November could even exceed those of summer. At Marambio, in some years, also dailydoses in October can be as high as those during the summer. At Ushuaia, the peak daily maximum UV index of 13 was measured twice: in November 2003 and 2009. Also during those days, the station of Ushuaia was inside the polar vortex.

*Copyright statement.* TEXT

## 1   Introduction

15   The NILU-UV Antarctic network was established in 1999/2000 in scientific cooperation between the Spanish Agencia Estatal de Meteorología (AEMET), the Finnish Meteorological Institute (FMI) and the Argentinian Dirección Nacional del Antártico-Instituto Antártico Argentino (DNA-IAA) and Centro Austral de Investigaciones Cientificas (CADIC). The goal was to promote



research of stratospheric ozone and ultraviolet (UV) radiation in the Antarctic region and to serve multidisciplinary research by providing UV and ozone data.

The well-known Antarctic ozone depletion was first discovered by Farman et al. (1985) in the mid-1980s. Satellite measurements confirmed that the ozone hole existed over whole Antarctica. Soon after, more in situ measurements were started to

monitor total ozone and UV radiation. Also regular balloon soundings were launched to monitor the profile of the ozone distribution in the atmosphere (e.g. Karhu et al. (2003)). The knowledge in the mechanisms of the ozone depletion increased, and the role of ozone depleting subtances and cold temperatures inside the polar vortex was understood (WMO, 1990, 1999; Pazmiño et al., 2005). Therefore the stations of the Antarctic NILU-UV network were chosen regarding their location: Belgrano (77°, 35°W) was mostly located inside, Marambio (64°S, 56°W) at the edge and Ushuaia (54°S, 68°W) outside the polar vortex.

NILU-UV multichannel radiometers (Høiskar et al., 2003) were set up at the stations to continuosly monitor total ozone and UV irradiance.

The measurements of the network were interrupted in 2014, and the whole UV time series of Ushuaia and Marambio are analysed for the first time in this study. Since the begining of the network, special attention was given to the quality assurance of the measurements in order to make measurements world wide comparable (Lakkala et al., 2005). A traveling reference was

used to transfer the irradiance scale to both Marambio and Ushuaia. As the instrument of Belgrano didn't follow the same quality assurance procedures, albeit it was regularly exchanged with a newly calibrated instrument, the time series of Belgrano is not discussed in this study.

The first measurements of the network started in the middle of the ozone depletion of the Antarctic ozone layer (WMO, 2003, 2007). Later, the recovery of stratospheric ozone was predicted (Andrady et al., 2009), and nowadays, 30 years after

signing the Montreal Protocol restricting ozone depleting substances, first signs of the stratospheric ozone recovery have been observed (Solomon et al., 2016). The NILU-UV time series of Ushuaia and Marambio presented in this study are important for the future: The time series serve as reference representing times when severe stratospheric ozone depletion occurred.

Observations at these locations complement measurements with spectroradiometers at the South Pole, McMurdo Station (78°S), Palmer Station (65°S) and Ushuaia, which started in the early 1990s and are part of the U.S. National Science Foun-

dation's (NSF) UV Monitoring Network (Booth et al., 1994; Bernhard et al., 2010). Bernhard et al. (2008) and Bernhard et al. (2010) analyzed the data of these sites and found a large effect of the ozone hole on the UV index at the three Antarctic sites, and to a lesser extent at Ushuaia. Their analysis showed that at South Pole the UV index is on average 20-80% higher during the ozone hole period than between January and March. They studied the UV index compared to estimates of UV index prior the ozone hole period, and found that the average UV index between 1991 and 2006 was 55%–85% higher than the estimates

for 1963–1980. The highest UV index measured at Palmer was 14.8. At Ushuaia, UV indices as high as 11.5 were measured in October, when the ozone hole was over the city. Eleftheratos et al. (2015) studied changes in irradiances at 305 and 325 nm at Southern latitudes 55°S–77°S, but didn't find any statistically significant changes between 1990 and 2011.



## 2  Materials and Methods

### 2.1  NILU-UV radiometers at Ushuaia and Marambio

The NILU-UV multichannel radiometer measures UV radiation with five channels with center wavelengths at 305, 312, 320, 340 and 380 nm. The full width half maximum (FWHM) of the measured wavelength band is around 10 nm. A sixth channel
measures the photosynthetically active radiation (PAR) at wavelenghs between 400 and 700 nm. Erythemally weighted irradiances, UVB (280–320 nm) and UVA (320–400 nm) irradiances can be retrieved from the measurements. In addition to UV irradiances, total ozone and information about the cloud optical depth can be calculated.

The instrument is described in details in Høiskar et al. (2003). It is weatherproof and designed to operate in harsh environment. The instrument has a flat Teflon diffuser, interference filters and silicon detectors. In order to keep the instrument stable
at different temperatures, internal heating keep the temperature constant at 40°C.

In routine operation, one minute averages are recorded by a built-in data logger. There is a possibility to record measurements each second, and this option was used during lamp measurements. Three weeks of one minute averages can be stored in the memory of the instrument. In the NILU-UV network, the data was however daily/weekly transfered automatically or manually to the servers of AEMET and FMI.

Ushuaia is located in the archipelago of Terra del Fuego, Argentina. The NILU-UV instrument was set up on the roof of the CADIC facility next to a SUV-100 spectroradiometer that was part of the U.S. National Science Foundation's Ultraviolet Spectral irradiance Monitoring Network (Booth et al., 1994). The altitude of the station is 23 m. The station is surrounded by mountains, which have snow cover most of the year. In the South, there is the Beagle Channel, which connects Ushuaia to the South Pacific Ocean. There is snow on the ground from June to September.

The research station of Marambio, Base Marambio II, is located in the point of the North East Antarctic penninsula. At Marambio, the NILU-UV was set up on the roof of the main scientific building. The altitude of the station is 196 m. The climate is characterized by mean temperatures below zero the whole year and strong winds. The temperatures range from -15°C in June to -2° in December (from http://www.smn.gov.ar/serviciosclimaticos/). Fog is frequent, and the most foggy months are December and January. Summer is the cloudiest season and winter have the least clouds. The snow stays on the
ground the year round.

### 2.1.1  Erythemally weighted UV products

The time series of erythemally weighted UV irradiances were calculated using the method described in Dahlback (1996). In this method, UV dose rates (D) are calculated from the measurements of the NILU-UV's five UV channels using a linear function (eq. 1). In the linear function, the raw data (V) measured at each channel (i) was multiplied with a coefficient (a),
which was determined during the calibration of the NILU-UV. The coefficients ($a_i$) of eq. 1 were determined by utilizing a combination of comparisons with a reference spectroradiometer and radiative transfer calculations. The spectroradiometer of the Norwegian Radiation Protection Authority (NRPA) was used as reference spectroradiometer (Johnsen et al., 2002). The action spectrum of McKinlay and Diffey (1987) was used, when calculating the coefficients to retrieve erythemally weighted





UV dose rates.

$$D = \sum_{i=1}^{5} a_i V_i \qquad (1)$$

In this work the UV indices and daily doses, calculated from the erythemally weighted dose rates, were studied. The UV index was calculated by multiplying the erythemally weighted dose rate expressed in unit $W/m^2$ by 40 (e.g., WMO (1997)). The erythemally weighted daily doses (H) were calculated by integrating the erythemally weighted dose rates over the whole day (eq. 2, $T_1$=00:00 h and $T_2$=23:59 h local time).

$$H = \int_{T_1}^{T_2} D(T)dT \qquad (2)$$

For each day of the year, daily means of daily maximum UV index and daily dose were calulated using the whole time series. From now on in this paper, these means are called "daily climatology", even if they are not real climatological means from statistical point of view as the time period is too short. Daily deviations (DEV) as percentage were calulated from these means following eq. 3, where X is the daily maximum UV index or daily dose and "mean" is the mean value (2000–2010) of that day.

$$DEV = \frac{(X - mean)}{mean} * 100\% \qquad (3)$$

## 2.2 QC of UV data at Marambio and Ushuaia

The quality control of the NILU-UVs of Marambio and Ushuaia were performed by the operators at the station. The everyday housekeeping included cleaning of the diffuser, check of leveling and clock adjustment. Regular lamp measurements were performed at the stations and the data was regularly delivered. The quality control of the NILU-UV network is explained in details in Lakkala et al. (2005) and shortly explained in the next section.

### 2.2.1 Lamp measurements

Lamp measurements were performed at the station every two weeks. A 100 W lamp was mounted on a lamp holder and placed over the NILU-UV. After the warm up of the lamp, the irradiance was measured during 15 minutes with a time step of one second. The temperature of the instrument was also recorded. Each time two lamps were measured and every third time a third lamp was measured in addition. This led to different burning times of the lamps, which made it possible to detect drifts in lamp irradiances.

Each lamp test was compared to the average of the first three measurements at each channels (Fig. 1 and 2). Severe drift in the signal of the instrument was already detected in channels 1–4 after a couple of years of operation of the NILU-UV. The large drifts were caused by the material from which the filters were made: They degraded due to UV radiation. After 4 years of operation, decreases up to 40% in the sensitivity of some filters were observed (Lakkala et al., 2005). The degradation continued through the whole measurement time period. After 2010, the sensitivity of the channels 2–4 decreased more than




80% (Fig. 1 and 2). Due to these large changes, it was no longer possible to correct data after 2010. Until that year, the data was corrected using the scaling of the channels to those of the traveling reference during solar comparisons (see Chapter 2.3.2). The data of the years 2011–2013 had to be excluded from the analysis.

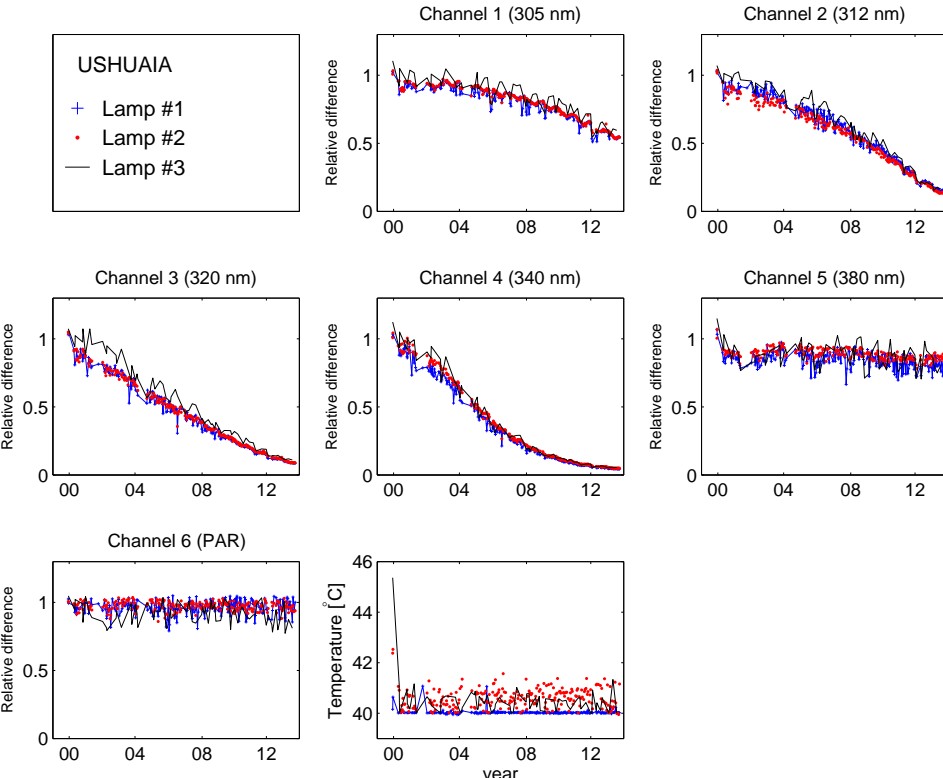

**Figure 1.** Lamp measurements at Ushuaia 2000-2013. The raw data of the lamp is divided by the mean of the three first mesurements. The temperature of the NILU-UV during the measurement is also shown.





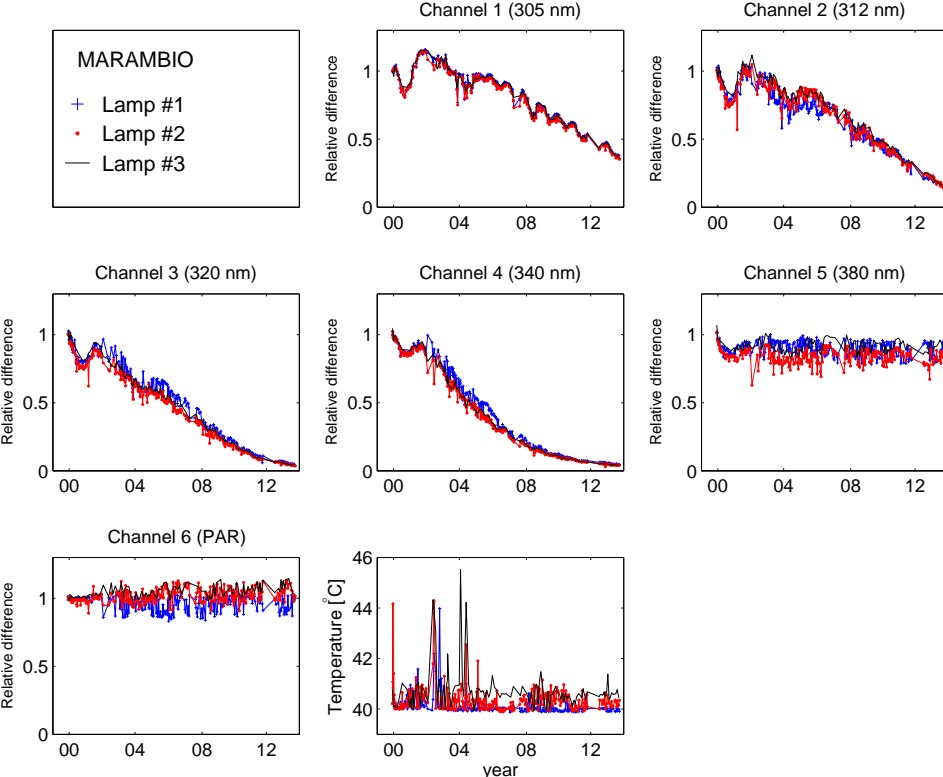

**Figure 2.** Lamp measurements at Marambio 2000-2013. The raw data of the lamp is divided by the mean of the three first mesurements.The temperature of the NILU-UV during the measurement is also shown.





### 2.3 QA of the network

The quality assurance of the network was based on the use of a traveling reference to transfer the irradiance scale to the NILU-UV radiometers of Ushuaia and Marambio. The details are explained in Lakkala et al. (2005) and the procedures are shortly described in the next sections.

### 2.3.1 Traveling reference

The traveling reference of the network was a NILU-UV radiometer, which had filters with similar spectral responses as the NILU-UV radiometers of Ushuaia and Marambio. It was calibrated by the manufacturer in Norway in 1999, 2000, 2001, 2002, 2004, 2005, 2007 and 2010. The irradiance scale was traceable to the National Institute of Standards and Technology (NIST) via the Swedish Testing and Research Institute (SP) (Johnsen et al., 2002).

The quality control of the traveling reference included lamp tests before and after each solar comparison. The quality assurance included comparisons with spectroradiometers in Finland and Ushuaia. In Finland, the traveling reference was compared to the Brewer spectroradiometers of FMI at either Sodankylä (67°N) or Jokioinen(61°N). The irradiance scale of the Brewer was traceable via the National Standard Laboratory MIKES, Aalto University, Finland, to SP (Heikkilä et al.; Lakkala et al., 2008). At Ushuaia, measurements were compared to data of the SUV-100 spectroradiometer of the NSF UV monitoring network. This network also included high-latitude sites in Antarctica and the Arctic (Bernhard et al., 2010). Measurements at Ushuaia were discontinued in December 2008. The irradiance scale was traceable to NIST (Bernhard et al., 2003).

In this study, results of the comparisons done in 2004–2012 were analysed for clear sky conditions (Tables 1 and 2). Results prior to 2004 were analysed and discussed in Lakkala et al. (2005). All comparisons showed that the difference between the traveling reference and the spectroradiometers was within 5% until February 2008. In 2009, the comparison against the Finnish Brewer spectroradiometer in Jokioinen showed a difference of 20% in the erythemally weighted dose rates. In 2010, the NILU-UV reference radiometer was sent to Norway to be recalibrated. The next comparisons were in 2011 and 2012, when the differences were 8% and 26%, correspondingly. After 2011, the channels of the traveling reference had drifted so severely that even recalibration would not have helped, and measurements with that instrument were not possible after 2012.





**Table 1.** The ratios of erythemally-weighted UV dose rates between the reference NILU-UV and the Brewer spectroradiometers #037 and #107 during 2004–2012.

| Date | SZA | Brewer#107/NILU-UV | Brewer#037/NILU-UV |
|---|---|---|---|
| 16.7.04 | 41 | 1.04 | |
| 18.8.05 | 48 | 1.01 | |
| 10.6.07 | 44 | | 1.03 |
| 16.6.07 | 37 | 1.02 | |
| 31.8.08 | 54 | 1.04 | |
| 6.8.09 | 45 | 0.80 | |
| 17.8.11 | 48 | 1.08 | |
| 19.7.12 | 48 | 1.26 | |

**Table 2.** The ratios of erythemally-weighted UV dose rates between the reference NILU-UV and the SUV-spectroradiometer during 2004–2006. For 2007–2008 the comparisons were done using the NILU-UV of Ushuaia.

| Date | SZA | SUV/NILU-UV |
|---|---|---|
| 16.1.04 | 52 | 0.95 |
| 10.12.04 | 40 | 0.96 |
| 1.3.05 | 48 | 1.03 |
| 28.11.05 | 38 | 0.96 |
| 3.3.06 | 48 | 0.97 |
| 15.11.06 | 40 | 0.99 |
| 12.2.07 | 44 | 1.00 |
| 26.10.07 | 44 | 1.02 |
| 2.2.08 | 41 | 1.00 |



**Table 3.** Number of good measurement days during 2000-2010.

| Year | Ushuaia | Marambio |
| --- | --- | --- |
| 2000 | 326 | 294 |
| 2001 | 327 | 327 |
| 2002 | 311 | 306 |
| 2003 | 289 | 317 |
| 2004 | 355 | 345 |
| 2005 | 338 | 338 |
| 2006 | 326 | 325 |
| 2007 | 330 | 346 |
| 2008 | 342 | 343 |
| 2009 | 339 | 335 |
| 2010 | 340 | 340 |

### 2.3.2  Solar comparisons to transfer the irradiance scale

The traveling reference typically visited Marambio twice and Ushuaia three times during spring–autumn. During these site visits, solar comparisons were performed to transfer the irradiance scale. Each channel of the site radiometer was scaled to the corresponding channel of the traveling reference. Thus, the calibration of the traveling reference could be used. The method is described in details in Lakkala et al. (2005). In the method, the calibration of the site instruments relevant for solar data was entirely based on the comparisons with the reference instrument and not the lamp scans described earlier.

The scaling coefficient time series is shown in Fig. 3. The differences between the channels 1–3 of the traveling reference and the site NILU-UV were less or around 50% until 2010 for both instruments. The channel 2 of the instrument of Marambio had however smaller differences than the corresponding channel of the instrument of Ushuaia. The channel four, which degraded the fastest, had more than 80% of difference already in 2008.

Until 2008, all measurements at Ushuaia and Marambio were scaled according to the method by (Lakkala et al., 2005) and the erythemally weighted UV dose rates were calculated. In 2009, following the results of solar comparison between the FMI Brewer spectroradiometer and the travelling reference, which showed a difference of 20%, all dose rates referenced to the travelling reference were scaled by 20% to account for the drift of this system. In 2010, the reference radiometer was recalibrated by the manufacturer, and the measurements at Ushuaia and Marambio were adjusted to match the measurements of this radiometer. The reference radiometer was again compared with the FMI Brewer spectroradiometer in 2011 and UV dose rates measured by the two systems agreed to within 8%. The number of days with good measurement that were used in this study, are shown in Table 3 for each year.





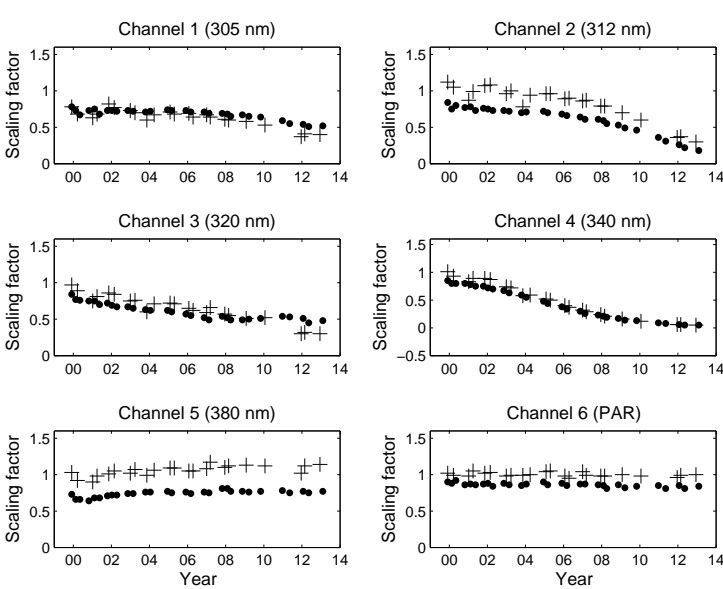

**Figure 3.** Scaling factors of the UV channels of the Ushuaia (.) and Marambio (+) NILU-UV instruments.





## 2.4 Potential vorticity analysis

Both Ushuaia and Marambio can be located inside, at the edge or outside the polar vortex. The impact of the location with respect to the polar vortex was studied by calculating the potential vorticity (PV) at Ushuaia and Marambio for measurement days. The modified PV, in which the PV was scaled to 475 K (Lait, 1994) using the ERA interim data, was used. If the PV was

higher than the value 36 (in South hemisphere lower than -36), the station could be classified to be inside the vortex.

## 3 Results and Discussion

After the quality assurance procedures mentioned in the earlier sections, the time series of erythemally weighted UV irradiances for Ushuaia and Marambio were calculated for the time period 2000-2010. Maximum daily UV indices and daily doses calculated from the results are presented in Figs. 4 and 5 and discussed in the next sections. The data is archived and can be

downloaded at the FMI Arctic Space Centre open access database http://litdb.fmi.fi.

### 3.1 UV index time series

At both stations, the highest UV indices were measured between the end of October and the end of November. As this period is before the summer solstice, factors other than the SZA must be responsible for the observed time-shift of the maximum UV index: The well-known Antarctic ozone depletion occurs during October-November (WMO, 2003) and the snow that still

might be on the ground during these months. Both factors increase UV irradiances near the ground. Furthermore, clear skies or broken cloudiness conditions occurred during days when the highest UV values were measured.

     At Marambio, extreme (i.e., UV indices of 11 or more, WMO (1997)) daily maximum UV indices were measured on 5 days, and the peak value of 12 was measured on 18 Nov 2007. The total ozone amount on that day was 213 DU. The daily maximum UV was very high (i.e., 8-10) on 85 days and high (i.e, 6-7) on 524 days. The daily climatological means calculated from

the daily maximum UV indices (2000–2010) were highest, UV index of 6, in late November, December and January (Fig. 6). Fig. 6 shows that the daily climatological means were not symmetrically distributed around mid summer, but increased in October compared to days with the same noon-time SZA in autumn. Peak UV indices occurred between late October and late November, depending on how long the polar vortex stayed above the station. That was also the period during which the maximum UV indices deviated the most from the daily means (Fig. 7). Also snow typically enhanced UV radiation in spring

(Meinander et al., 2014). Measured mid summer clear sky daily maximum UV indices were between 7 and 8. In June, no UV radiation was measured at the station. For comparison: At the same latitude in Nothern hemisphere, in Finland, the highest measured UV indices were between 5 and 6. The main reasons for higher summer UV indices in Marambio were lower total ozone amounts and shorter Earth-Sun distance.

     At Ushuaia, extreme daily maximum UV index was measured on 35 days. The highest value, UV index 13, was measured

twice: on 27 Nov 2003 and on 15 Nov 2009. The corresponding total ozone amount of 27 Nov 2003 was 268 DU, but no ozone data was recorded for 15 Nov 2009. However, low ozone was evident as the total ozone amounts of the previous and





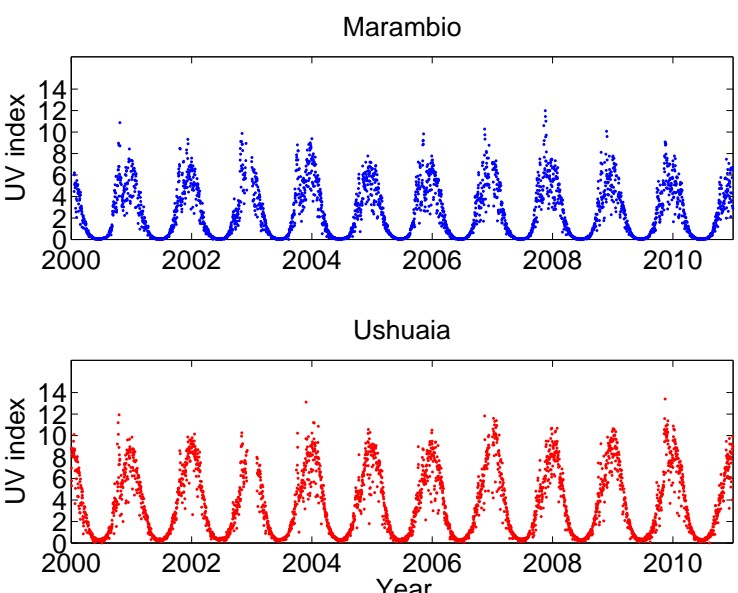

**Figure 4.** Maximum UV index measured at Marambio and Ushuaia 2000–2010.

next days were 176 and 215 DU, respectively. Very high daily maximum UV index was measured on 515 days and high on 1045 days. Daily climatological means calculated from the daily maximum UV indices 2000–2010 were highest, UV index 8–9, in January (Fig. 6). The graph of the daily climatological mean showed the same, but not as pronounced, features than at Marambio: Enhanced daily maximum UV indices in October (Fig. 7). When the station was inside the polar vortex in October,

5   the maximum UV indices could be as high as those measured in the summer, when they were between 10 and 11.





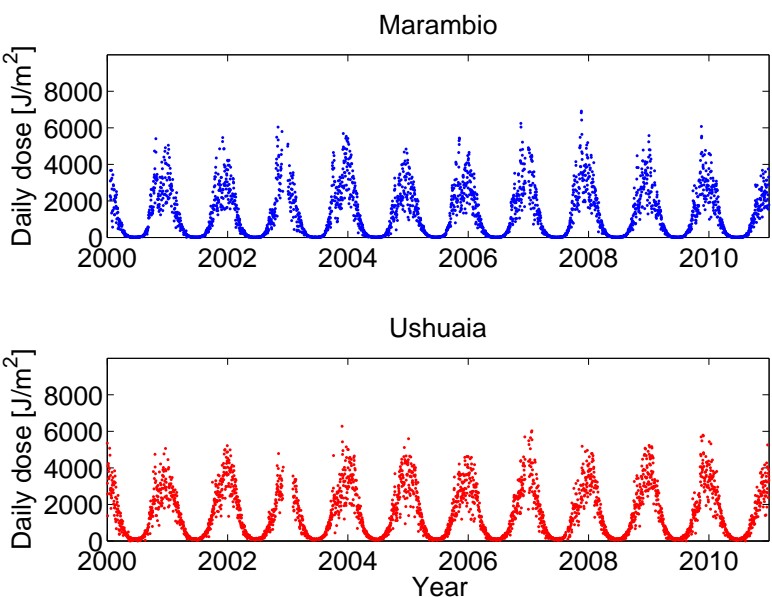

**Figure 5.** Erythemally weighted daily doses measured at Marambio and Ushuaia 2000–2010.

The deviations from the daily climatological mean were calculated for each day (Fig. 8) and analyzed with respect to the location of the polar vortex. All deviations larger than 80% from the mean occurred when the stations were inside the polar vortex. At Ushuaia this was the case especially in 2000–2004 and in 2009. At Marambio, deviations of over 100% were measured in 2000, 2002, 2003 and 2007. Highest deviations were 122% on 1 Oct 2003 and 145% on 12 Oct 2000 at Marambio and Ushuaia, respectively. During cloudy days, the daily maximum UV index did not reached high values, even if the station





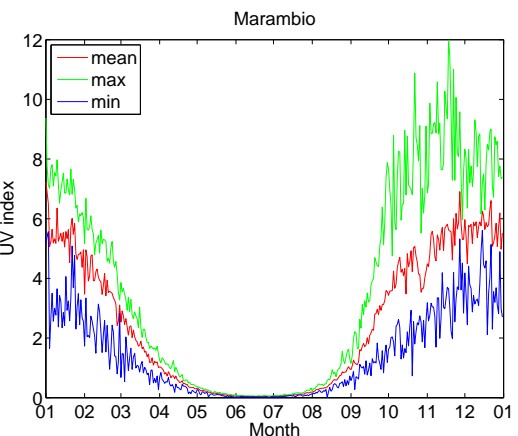
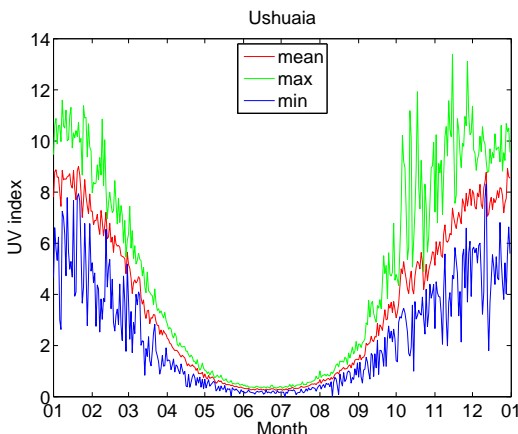

(a)                                                      (b)

**Figure 6.** The average, maximum and minimum values of the daily maximum UV index calculated for each day of the year in the period 2000–2010 in a) Marambio b) Ushuaia.

was inside the polar vortex. Lowest values occured during days with heavy cloudiness or rainy days, with deviations of 100% from the daily mean.




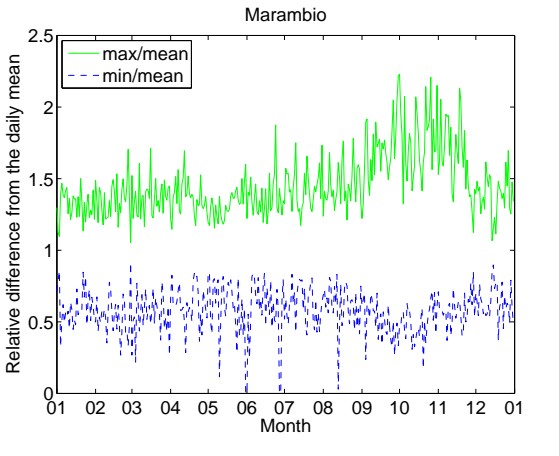

(a)

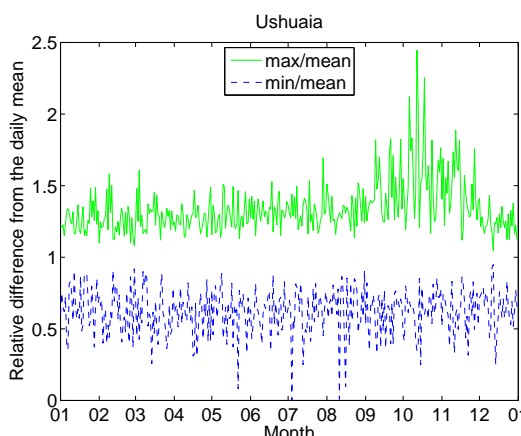

(b)

**Figure 7.** The maximum and minimum values of the daily maximum UV index divided by the mean of maximum UV index on that day in a) Marambio b) Ushuaia 2000-2010.





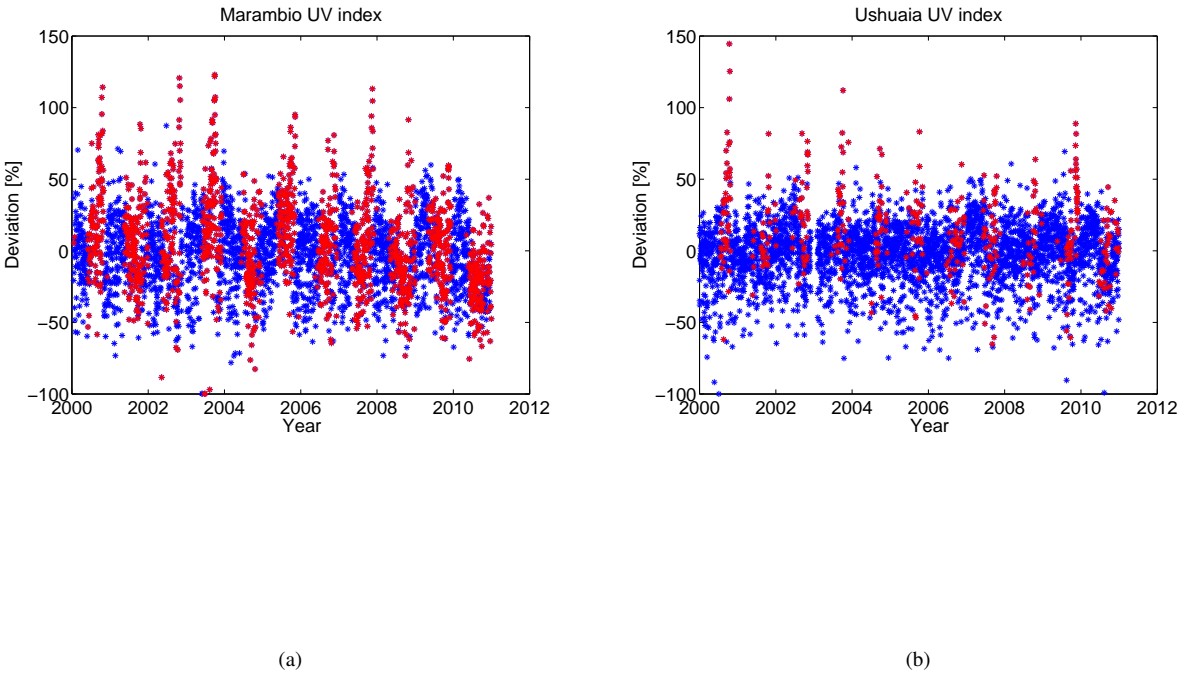

(a)                                                    (b)

**Figure 8.** Deviation from the daily mean maximum UV index in a) Marambio and b)Ushuaia for the years 2000-2010. Days for which the station was inside the polar vortex are indicated with a red symbols, the others in blue.



## 3.2 UV daily dose time series

Daily doses were strongly dependent on the total ozone amount and cloudiness conditions. For some days with low total ozone and clear sky, the daily dose time series graph showed that peak daily doses were more pronounced than peak UV indices (Fig. 4). The highest daily doses were around 6 $kJ/m^2$, at both Marambio and Ushuaia , even though the difference in latitude

between these sites is $10°$. At Marambio, a daily dose of 7 $kJ/m^2$ was measured in 2007, when the station was inside the polar vortex and episodes with low total ozone occurred. The highest daily doses were measured at Ushuaia and Marambio on 27 Nov 2003 and 19 Nov 2007, respectively.

The graph of the daily climatological mean of the daily doses (Figs. 9 and 10) showed features similar to those of the daily climatological mean of maximum UV indices (Fig. 6). The impact of low ozone episodes in October is seen, but not as clearly

as for daily maximum UV indices. At Marambio, the highest mean daily doses occurred between late November and the end of December. When the station was inside the polar vortex and clear sky conditions occurred, the typical daily dose was around 5 $kJ/m^2$ in November. The typical daily dose was between 2.5 and 3.5 $kJ/m^2$ in summer, but around 4.5 $kJ/m^2$ on sunny days. The high fluctuation in maximum values is due to the impact of year to year changes in cloudiness and total ozone.

At Ushuaia, typically daily doses were between 3 and 4 $kJ/m^2$ in summer. The graph of the climatological daily mean of

daily doses showed maxima between late December and the end of January. Highest daily doses, between 4.5 and 5 $kJ/m^2$, occurred typically in late January. However, daily doses of the same amount were measured on days with clear sky and low total ozone in late November. Compared to Marambio, the period of highest daily doses was shorter. At Marambio the period of highest daily doses, over 4 $kJ/m^2$, started already in the beginning of October, while it started at Ushuaia in November. At Marambio, in addition to more frequent low total ozone conditions, also the snow on the ground increased the UV irradiances

due to high surface reflection of the incoming radiation.

Both stations were inside the polar vortex on most of the days when the daily dose deviated by more than 75% from the daily climatological mean (Fig. 11). At Ushuaia this was the case in 2000 – 2003. The highest deviation of 141% from the mean was measured on 6 Oct 2003. At Marambio, in all cases except one, the station was inside the polar vortex when the daily doses deviated by more than 100% from the climatological mean of the day. The highest deviation of 150% from the

daily climatological mean was measured on 3 Nov 2002. Deviations exceeded 100% from the daily mean also in 2000, 2003, 2005, 2007 and 2008.





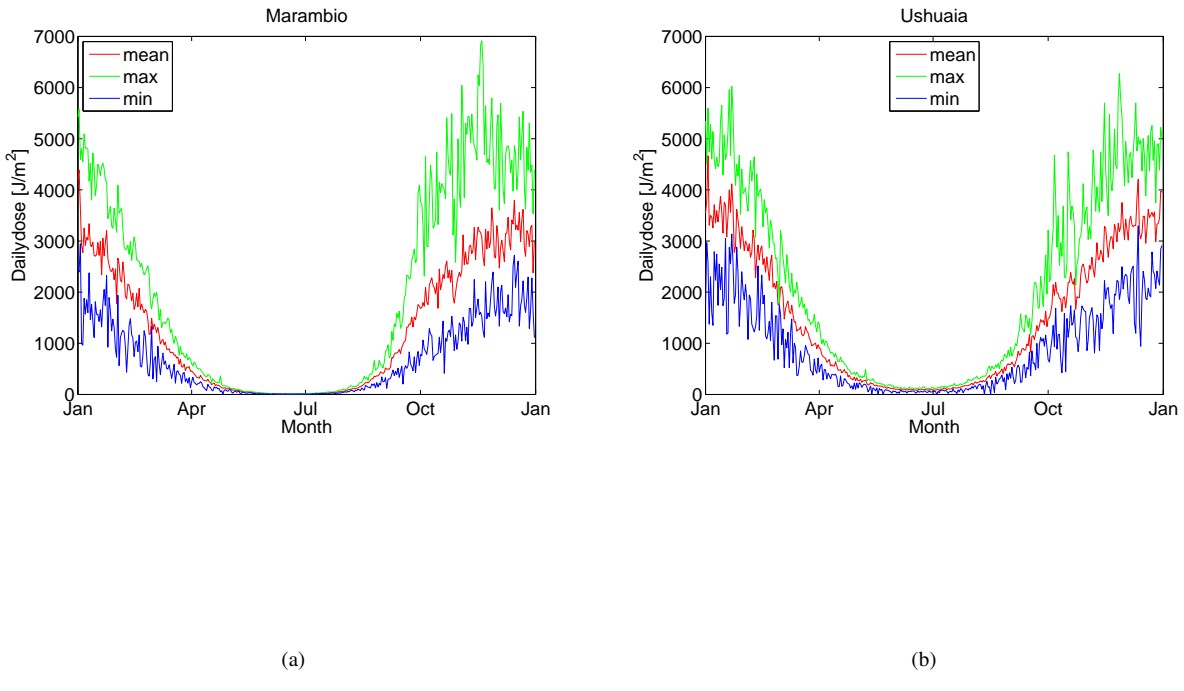

(a)                                      (b)

**Figure 9.** The mean, maximum and minimum calculated for each day of the year from the erythemally weighted dailydoses $[J/m^2]$ measured in a) Marambio b) Ushuaia 2000-2010.



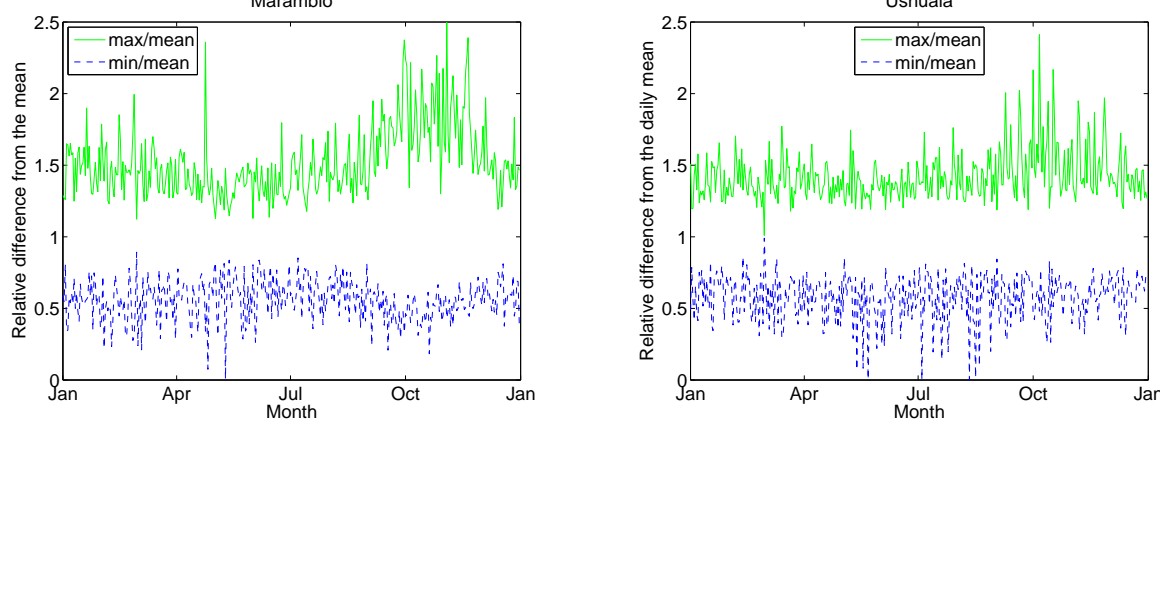

**Figure 10.** The maximum and minimum calculated for each day of the year from the erythemally weighted dailydoses divided by the mean of the same day in a) Marambio b) Ushuaia 2000-2010.





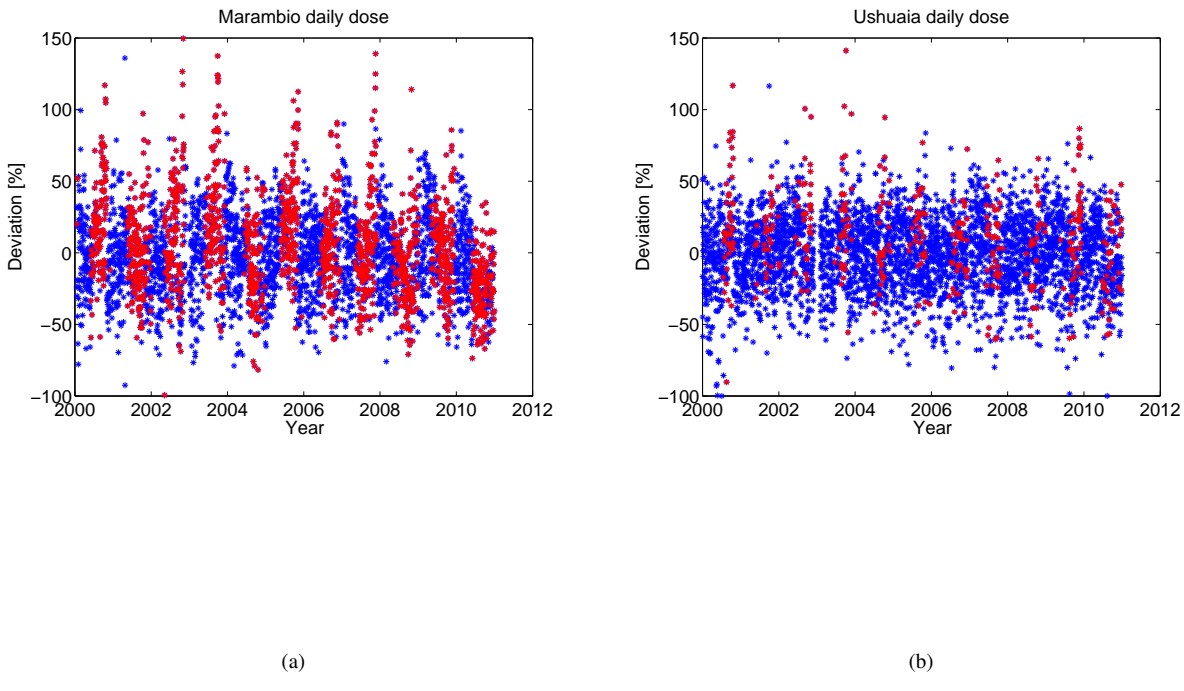

(a)  (b)

**Figure 11.** Deviation from the daily mean erythemal daily doses in a) Marambio and b) Ushuaia for the years 2000-2010. Days for which the station was inside the polar vortex are indicated with a red symbol, the others in blue.





## 4 Future

After the operation of the Antarctic NILU-UV network until 2014, co-operative Argentinian-Finnish UV measurements were continued using erythemally weighted UV sensors. In cooperation between FMI and Servicio Meteorológico Nacional (SMN), Argentina, UV broadband radiometers, type SL501A manufactured by Solar Light Co., were installed in 4 February 2013 to

measure the local bihemispherical UV albedo (Meinander et al., 2014). One instrument was installed to measure UV radiation reflected from the ground, and a second one to measure global UV irradiance. Using these measurements, the UV index time series could be continued, as those measurement are also traceable, via the primary calibration lamp, to the National Standard Laboratory MIKES, Aalto University, Finland. The official SL501A trace is to the primary standard of the National Institute of Standards and Technology (NIST). The difference in calibration coefficients using NIST and MIKES has been found to be less

than 2%, and in comparison of spectral UV irradiance scales maintained by NIST, PTB and Aalto University (HUT), no major differences have been found (Jokela et al., 2000). Hence, the irradiance scale of these new SL501A radiometers is comparable to that of the NILU-UV measurements within 2%.

Likewise in the scientific cooperation between SMN and FMI, a new GUV-2511 multi-filter radiometer, manufactured by Biospherical Instruments Inc., was installed at Marambio in 2017. It features five channels at UV wavelengths and two chan-

nels at visible radiation wavelengths (Bernhard et al., 2005). Two GUV radiometers were bought by FMI to ensure continuous measurements at Marambio: while one of them is installed at the site, the other one is calibrated and also available for comparison campaigns. These measurements continue the NILU-UV time series, as the central wavelengths of the channels are nearly identical to those of the NILU-UV channels and the bandwidths are similar. The first calibration of the GUVs were performed by Biospherical Instruments, Inc. Comparisons with FMI's spectroradiometers showed that the measurements were within 5%

from each other. These results are promising for the future of the UV time series at Marambio, and for monitoring the impact of the predicted ozone layer recovery on surface UV radiation in the Antarctic. The measurements will also serve as important validation data for satellite measurements, similar to the the NILU-UV measurements previously.

## 5 Conclusions

In this paper, UV measurements from the Antarctic NILU-UV network at Marambio (64°S) and Ushuaia (54°S) were quality

controlled and analyzed. The studied time series included measurements from 2000 to 2013. The quality assurance procedures included in situ lamp measurements and solar comparisons with the traveling reference of the network.

The results showed that after the year 2010 the filters of the radiometers degraded and their sensitivity to UV radiation was reduced by more than 80% in some channels. Due to this problem, accurate data were available only until 2010. The irradiances measured by the traveling reference were compared to those of well maintained spectroradiometers in Finland and

Argentina. From these results, also the traveling reference data were found to be accurate only until 2010. The data of the stations of Ushuaia and Marambio were scaled to that of the traveling reference and homogenized UV irradiance time series were calculated.



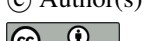

In both Ushuaia's and Marambio's time series, the effect of low total ozone amount during days that the station was inside the polar vortex was clearly seen. At Marambio, daily doses for late October and November could even exceed those measured in the summer. The maximum daily dose was as large as 7 $kJ/m^2$ in November 2007. Highest daily doses were typically around 6 $kJ/m^2$ at both stations, even if the difference in latitude is 10°. In the whole measurement period (2000–2010) there

were 5 days when extreme UV indices and 85 days when very high UV indices were measured at Marambio. The highest daily maximum UV index of 12 was measured in November 2007. At Ushuaia, extreme UV indices were measured on 35 days. The highest UV index of 13 was measured twice: in November 2003 and 2009. Largest deviations from the daily means were over 100% and occured in October and November when the stations were inside the polar vortex. At Ushuaia, the largest deviations from the mean occured during the first years of the time series, i.e, 2000–2003.

*Code availability.* TEXT

*Data availability.* The data can be downloaded at the FMI Arctic Space Centre open access database http://litdb.fmi.fi.

*Code and data availability.* TEXT

*Author contributions.* K. Lakkala: Primarily responsible for the QA of the UV data, analysed the data and led the manuscript preparation.

A. Redondas: Principal Investigator on the Antarctic NILU-UV project and contributed to the manuscript.

O. Meinander: QC of the traveling reference. One of the main authors of the FARPOCC and SAARA research plans of PI prof. Esko Kyrö.

L. Thölix: Analyzed the polar vortex and contributed to the manuscript.

B. Hamari: QA of the UV data 2007-2013.

F. Almansa: QA of NILU-UV Antarctic network.

V. Carreno: QA of NILU-UV Antarctic network.

G. Deferrari: Daily housekeeping and QC of NILU-UV data at Ushuaia.

H. Ochoa: Daily housekeeping and QC of NILU-UV data at Marambio.

G. Bernhard: QA of the SUV spectroradiometer data at Ushuaia. Calibration of FMI's GUV radiometers. Contributed to the writing of the manuscript

R. Sanchez: QC of the GUV measurements at Marambio.

G. de Leeuw: Overseeing the work in the group and contributed to the writing of the manuscript.





*Competing interests.* No competing interests are present.

*Acknowledgements.* The Academy of Finland has given financial support for this work through projects FARPOCC and SAARA. The MAR Project was financed by the National R+D Plan of the Spanish Ministry of Science and Technology (National Research Program in the Antarctic) under contract REN2000-0245-C02-01. The SUV-100 spectroradiometer UV data from Ushuaia were provided by the NSF UV Monitoring Network, operated by Biospherical Instruments Inc. and funded by the U.S. National Science Foundation's Office of Polar Programs. We thank the operators of the MAR project at Marambio and Ushuaia. Arne Dahlback and Björn Johnsen are acknowledged for the calibration of the NILU-UVs. The OMI/AURA and TOMS teams are thanked for the satellite ozone data. We thank Eija Asmi and Edith Rodriguez from FMI for logistics, and Lasse Ylianttila from the Finnish Radiation and Nucleation Safety Authority (STUK) for calibration of SL501A radiometers.





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
