# Peer review of "UV measurements at Marambio and Ushuaia during 2000-2010"

_Atmospheric Chemistry and Physics, 2017_

## Referee Comment (RC1) · Anonymous Referee #1 · 5 Mar 2018

The manuscript presents results of measurements of the UV index with multi-filter instruments at 2 sites in southern South America and Antarctica between 2000 and 2010. The results show clearly the strong effect of the Antarctic ozone hole on daily maximum levels and on daily totals. Very careful measures for QA/QC were applied to the data and are described in detail in the manuscript, because the sensitivity of the instruments was not stable. This leads to a final high quality dataset of daily values over 10 years, which is a significant contribution to our knowledge of distribution and variability of the levels of erythemally weighted UV irradiance at high southern latitudes. Therefore I think the manuscript is well suited to be published in ACP.

The manuscript is clearly structured and well written and the abstract gives relevant information. A few suggestions should be considered by the authors prior to publication:

[Figure]

The authors should give an estimate of the absolute uncertainty of the presented UV index data.

The authors state correctly that for clear sky conditions the solar zenith angle, the ozone amount and the coverage of the ground with snow are the most important parameters. To show the effect of ozone and snow more specifically, an additional figure with the time series of the UV index at a fixed solar zenith angle (e.g. 70° or 75°) could be impressive. Then, potentially, the relation between the UV index and ozone could be interpreted not only qualitatively, but also quantitatively (e.g. in terms of the radiation amplification factor).

I wonder that in Fig. 7a and 10a the variability of the ratios maximum/mean and minimum/mean is quite similar throughout the year, even on the days around winter solstice, when the absolute level of the mean is extremely small. Therefore very small differences at this time should give much higher variability of the ratio compared to the other days of the year.

Technical details:

Several times the term "daily mean" is used, but sometimes this is a bit misleading (e.g. p.4, l.8 or legend of Fig. 8). "Daily mean" should not be understood as the mean over the day, but as the day-to-day variation of the mean, where the mean is calculated from the respective days of each year of observations.

p.13, l. 5: "did not reach" instead of "did not reached".

p. 17, l. 8: "Fig. 9" instead of "Figs. 9 and 10".

---

## Referee Comment (RC2) · Anonymous Referee #2 · 27 Mar 2018

The manuscript of Lakkala et al. is sort of a follow-up paper of Lakkala et al. 2005 - "Quality assurance of the solar UV network in the Antarctic". It describes a 10-year long measurement series of UV measurements performed with multichannel radiometers. The instruments have been in continuous operation since 2000 at Marambio-Station (Antarctic peninsula) and Ushuaia (South Argentina). The authors give a well-rounded introduction into the ozone topic and the goals of the NILU-UV network. The introduction is followed up by a description of the radiometers and their corresponding measuring sites, as well as a description of how the erythemally weighted UV products presented in this work are calculated. The authors deliver a comprehensive description of the applied QC/QA procedures. The authors highlight UV index and UV dose time series and show the correlation of high UV occurrences while the stations are in-

side the polar vortex. All in all, these are well documented time series of data that are made available to the scientific community and should therefore be published in ACP.

I encourage the authors to consider the following comments:

p1, l. 7: "daily doses" instead of "dailydoses", pls change throughout the document

p2,l.13: "begining" -> "beginning"

Capitalize references to specific figures, tables, chapters, sections, equations: p3, l.29: "eq. 1" -> "Eq. 1". Not sure about the specific style guides of ACP, but I suggest to capitalize e.g. Equation 1, Figure 1, Appendix C since they are names of the entities they refer to. Please unify throughout the manuscript.

p4, l20ff: you present lamp measurements in great detail just to inform the reader four sites later on p9, l6. that they are not used for the instrument calibration. I recommend giving that information already in Section 2.2.1.

p9, l2: You state that the instruments are calibrated 2-3 times a year, some channels show a significant drift during that time scale. How big are the steps that occur in your data after those calibrations? Provide a percentage (up to x

p9, l8-10 channel 2 -> channel two

Results: Your results focus on the presentation of long-term time series and daily means - did you consider mentioning the number of measurements performed per day (or the time intervals between single measurements)? You could for example show data of one particular day to illustrate the capabilities of tracking fast ozone changes. If the paper gets too lengthy consider skipping some figures of UV daily dose since they show a very similar behavior like those of the UV index.

―――――――――――――

---

## Editor Comment (EC1) · A.F. Bais (Editor) · 2 Jul 2018

Figures 1 and 2: Please revise the layout of panels. You could a layout similar to that of Fig. 3 for the six channels and a fourth row showing the temperature and legend.

Figure 4: Change caption to: Daily maximum UV index . . .. Vertical axes labels are too dense. Either increase the axes height, or reduce the number of labels.

Figure 5. You may consider changing the data to kJ/m2; the vertical axes labels with thousands look quite ugly. Same for Figure 11.

Figures 7 and 8: Labels are too small.

All Figures: Please maintain consistency in the size of labels for all figures.

---

## Author Response (AR1)

**Final Author comments**

Authors' response to Referee #1, Referee #2 and Editor's comments on "UV measurements at Marambio and Ushuaia during 2000-2010" by Kaisa Lakkala et al.

The authors thank the Referees and the Editor for constructive comments and reply to all comments here below. The answer is structured as follow:  (1) comments from Referee/Editor, (2) author's response, (3) author's changes in the manuscript.

**Referee #1**

(1) The authors should give an estimate of the absolute uncertainty of the presented UV index data.
(2) To estimate the uncertainty of the presented UV index data, we calculated the combined uncertainty and used the coverage factor of 2.
(3) The text has been updated to be at Page 10, lines 15-19: "Using the a step of 5% as uncertainty related to the drift of the calibration of the site instrument between calibrations, the drift of the traveling reference between calibrations (3% for 2004–2006), the uncertainty of the calibration of the traveling reference (Lakkala et al., 2005), the uncertainties related to the transfer of the irradiance scale (Lakkala et al., 2005) and the uncertainty of the method to calculated the dose rates (Dahlback, 1996), the combined uncertainty was calculated to be 9.5%. The expanded uncertainty was 19% using a coverage factor of 2 (Table 3).

(1) The authors state correctly that for clear sky conditions the solar zenith angle, the ozone amount and the coverage of the ground with snow are the most important pa-rameters. To show the effect of ozone and snow more specifically, an additional figure with the time series of the UV index at a fixed solar zenith angle (e.g. 70◦ or 75◦) could be impressive. Then, potentially, the relation between the UV index and ozone could be interpreted not only qualitatively, but also quantitatively (e.g. in terms of the radiation amplification factor).
(2) The UV indices at SZA between 69.95 and 70.05 were selected to study the impact of snow in Marambio. The impact of snow was not supposed to be clearly seen, as there is snow on the ground almost year round. The amount of snow may vary depending on the weather conditions, but the highest snow amount occurs during winter and the smallest amount during summer months. Clear days were selected. For the whole time series, the RAF was calculated using the power law and it was found to be 1.16. For biggest ozone deviations, this RAF underestimated the effect in UV radiation, where RAF of around 1.3 were seen. Biggest ozone variations occurred in October, when there most likely snow in the ground at the station of Marambio.
(3) The plot of the RAF was including in the manuscript together with the corresponding text:
P. 4: " The radiation amplification factor (RAF) was calculated for erythemally weighted UV irradiances measured at SZA between 69.95 and 70.05 in Marambio. Clear sky data was selected using the cloud transmission factor (CLT) calculated from the NILU- UV irradiance at 380 nm (Høiskar et al., 2003). The power law (e.g., Booth and Madronich (1994)) was used to calculate the RAF ( Eq. (5)).
RAF =ln(O 3 /O 3 ∗ )/ln(U V ∗ /U V ) (5)
where O 3 ∗ is the first total ozone value and O 3 the second, respectively. U V ∗ is the first erythemally weighted UV irradiance and U V the second, respectively.

P 13: "The RAF was calculated to be 1.16, when excluding the winter months May-July from the analysis. When largest ozone decreases occured, the RAF could exceed 1.25 (Fig. 9) suggesting an additional increase due to snow on the ground."

(1) I wonder that in Fig. 7a and 10a the variability of the ratios maximum/mean and min-imum/mean is quite similar throughout the year, even on the days around winter sol-stice, when the absolute level of the mean is extremely small. Therefore very small differences at this time should give much higher variability of the ratio compared to the other days of the year.
(2) The calculations were checked and no mistakes were found. We agree that the climatological mean of daily maximum UV index and daily dose is very small around the winter solstice, but also the variability is small during that period, as the sun is near the horizon (high SZA) and low intensity . The variability of the rations maximum/mean and minimum/mean ranged from 1.2 to 1.8 and from 0.001 to 0.85, respectively.  During the other days of the year, the variability can be as high, e.g. for rainy days, when the minimum is very low.
(3) No update was made to the text.

Technical details:
(1) Several times the term "daily mean" is used, but sometimes this is a bit misleading (e.g. p.4, l.8 or legend of Fig. 8). "Daily mean" should not be understood as the mean over the day, but as the day-to-day variation of the mean, where the mean is calculated from the respective days of each year of observations.
(2) The authors have checked the text regarding the use of the term " daily mean". And changed the wording to clarify the text.
(3) The following sentences have been updated: P4 l. 14-18"For each day of the year, means of daily maximum UV index and daily dose were calculated from the respective days of each year of observation. From now on in this paper, these means are called "daily climatology", even if they are not real climatological means from statistical point of view as the time period is too short. Daily deviations (DEV) as percentage were calculated from this daily climatology following Eq. (4), where X is the daily maximum UV index or daily dose and "clim" is the daily climatology."

(2) From that paragraph onwards, the "daily mean" which was calculated from the respective days of each year of observations was called "daily climatology".

(1) p.13, l. 5: "did not reach" instead of "did not reached".
(2,3) Changed.

(1) p. 17, l. 8: "Fig. 9" instead of "Figs. 9 and 10".
(2)  Changed.
(3) The reference to the Fig. 10 (now Fig. 11) is added to the following sentence: "The high fluctuation in maximum values (Fig. 11) is due to the impact of year to year changes in cloudiness and total ozone."

**Referee #2**

(1) p1, l. 7: "daily doses" instead of "dailydoses", pls change throughout the document

(2,3) Changed throughout the document.

(1) p2,l.13: "begining" -> "beginning"
(2,3) Changed.

(1) Capitalize references to specific figures, tables, chapters, sections, equations: p3, l.29: "eq. 1" -> "Eq. 1". Not sure about the specific style guides of ACP, but I suggest to capitalize e.g. Equation 1, Figure 1, Appendix C since they are names of the entities they refer to. Please unify throughout the manuscript.
(2,3) References to specific figures, tables, chapters, sections and equations have been capitalized and changed to follow the style guides of ACP.

(1) p4, l20ff: you present lamp measurements in great detail just to inform the reader four sites later on p9, l6. that they are not used for the instrument calibration. I recommend giving that information already in Section 2.2.1.
(2) The information has been moved earlier.
(3) The text has been updated and is now: Page 5, line 3 " The results of the lamp tests were used to confirm the studied changes in the scaling coefficients, but not used in the absolute calibration."

(1) p9, l2: You state that the instruments are calibrated 2-3 times a year, some channels show a significant drift during that time scale. How big are the steps that occur in your data after those calibrations? Provide a percentage (up to x
(2) The percentage has been provided and added to Page 10 line 14.
(3) The following text was added to the manuscripts: "The highest step between calibrations occured in Marambio in January 2000 and it was 11% for erythemally weighted UV dose rates."

(1) p9, l8-10 channel 2 -> channel two
(2,3) Changed.

(1) Results: Your results focus on the presentation of long-term time series and daily means - did you consider mentioning the number of measurements performed per day (or the time intervals between single measurements)? You could for example show data of one particular day to illustrate the capabilities of tracking fast ozone changes. If the paper gets too lengthy consider skipping some figures of UV daily dose since they show a very similar behavior like those of the UV index.
(2) The following sentence already describes the frequency of recored measurements per day: P3 l 11 " In routine operation, one minute averages are recorded by a built-in data logger." We added a plot (Figure 6) of the record high UV index day, 18.11.2007 in Marambio to illustrate the capabilities of tracking fast ozone/UV changes. And added the following sentence on P3 "This allows the detection of rapid changes in cloudiness and total ozone amounts."
(3) The text is now:
P3: "In routine operation, one minute averages are recorded by a built-in data logger. This allows the detection of rapid changes in cloudiness and total ozone amounts."
and
P14, l. 5: At Marambio, extreme (i.e., UV indices of 11 or more, WMO (1997)) daily maximum UV indices were measured on 5 days, and the peak value of 12 was measured on 18 Nov 2007. The daily variation of the UV index on that record day is shown in Fig. 6. The total ozone amount on that day was 213 DU.

**Editor**

Figures 1 and 2: Please revise the layout of panels. You could a layout similar to that of Fig. 3 for the six channels and a fourth row showing the temperature and legend.
(2,3) The Figures have been revised and updated as suggested by the Editor.

Figure 4: Change caption to: Daily maximum UV index. Vertical axes labels are too dense. Either increase the axes height, or reduce the number of labels.
(2,3) The caption has been changed and the axes height has been increased.

Figure 5. You may consider changing the data to kJ/m2; the vertical axes labels with thousands look quite ugly. Same for Figure 11.
(2,3) Changed to kJ/m^2

Figures 7 and 8: Labels are too small.
(2,3) Labels have been increased. The final size of the labels will depend on the size of the figure in the final version.

All Figures: Please maintain consistency in the size of labels for all figures.
(2,3) The figures have been harmonized regarding the labels.

Ricardo SanchezGerardus de Leeuw

[revised manuscript text omitted]

---

## Author Response (AR2)

**Authors' answer to Co-Editor's comments on manuscript "UV measurements at Marambio and Ushuaia during 2000–2010" by Lakkala et al.**

The authors thank the Co-Editor for constructive comments and respond to each comment here below.

**Co-Editor Decision: Publish subject to minor revisions (review by editor)** (13 Sep 2018) by Alkiviadis Bais
Comments to the Author:
Page 14, line 12 states: The RAF was calculated to be 1.16, when excluding the winter months May-July from the analysis. When largest ozone decreases occurred, the RAF could exceed 1.25 (Fig. 9).

However, Figure 9 shows the percentage change in UVI and total ozone and not RAF. How the quoted values of the RAF were calculated; as the average of all daily values for each set of months, or by fitting an exponential to the UV and ozone changes? In the second case you could print the fit curve on the graph to show the difference between the two groups of months.

Answer: The RAF was calculated by fitting a first degree polynomial function using the ordinary least squares line fitting method. The explanation of the calculation of RAF was modified and it is now:

P. 4, line 22: RAF values were calculated from the power law (e.g., Booth and Madronich (1994))
$$RAF = \ln(O_3/O_3*)/\ln(UV*/UV) \quad (5)$$
using a least square fit, where $O_3*$ is the highest total ozone value observed during the time series and $UV*$ is the corresponding erythemally weighted UV irradiance.

The Figure 9 has been updated to include the theoretical RAF curves for both groups of months.

The text of the Section 3.1, page 14 was changed to:
"The RAF was calculated to be 1.16, when excluding the winter months May-July from the analysis. When calculating the RAF from data of months with the highest snow amount (August, September, October, March and April) the RAF was 1.18 (Fig. 9). For summer months November-February the RAF was calculated to be 0.91."

Figures 10 and 13 are identical and are referenced in different sections of the text. Please remove one of them and change the reference to Fig. 13 in the text.

Answer: The Figures 10 and 13 are not identical as the Fig. (10) shows deviation of maximum UV index from the climatological mean and Fig. (13) deviation of daily dose. The authors think both figures are justified as they include interesting information. For example, at Ushuaia when comparing the years 2000 and 2003 at time the station was inside the polar vortex (red color): The maximum UV index deviated more from the climatological mean in 2000 than in 2003, for the daily dose it was the opposite.

Please check the language again, because there are still some typos and syntax errors.
Typos have been corrected and the language checked.

For example:
in caption of Fig 10. Remove "a" before red. Removed.

P22, L22: Remove second occurrence of "the" before "NILU-UV". Removed.
P23, L6. Replace occured with occurred. Replaced.

[revised manuscript text omitted]